# Protective Effect of Baicalin on Chlorpyrifos-Induced Liver Injury and Its Mechanism

**DOI:** 10.3390/molecules28237771

**Published:** 2023-11-25

**Authors:** Ruibing Wang, Ke Zhang, Kaiyue Liu, Hongyan Pei, Kun Shi, Zhongmei He, Ying Zong, Rui Du

**Affiliations:** 1College of Chinese Medicinal Materials, Jilin Agricultural University, Changchun 130118, China; 13159533668@163.com (R.W.); wojiaozhangkea@163.com (K.Z.); phy19990505@163.com (H.P.); sk1981521@jlau.edu.cn (K.S.); heather78@126.com (Z.H.); 2Jilin Provincial Engineering Research Center for Efficient Breeding and Product Development of Sika Deer, Jilin Agricultural University, Changchun 130118, China; 15870540735@163.com

**Keywords:** chlorpyrifos, baicalin, acute liver injury, autophagy, apoptosis

## Abstract

Chlorpyrifos (CPF) plays a vital role in the control of various pests in agriculture and household life, even though some studies have indicated that CPF residues pose a significant risk to human health. Baicalin (BA) is a flavonoid drug with an obvious effect on the prevention and treatment of liver diseases. In this study, the protective effect of BA in vitro and in vivo was investigated by establishing a CPF-induced AML12 cell damage model and a CPF-induced Kunming female mouse liver injury model. The AML12 cell damage model indicated that BA had a good positive regulatory effect on various inflammatory factors, redox indexes, and abnormal apoptosis factors induced by CPF. The liver injury model of female mice in Kunming showed that BA significantly improved the liver function indexes, inflammatory response, and fibrosis of mice. In addition, BA alleviated CPF-induced AML12 cell damage and Kunming female mouse liver injury by enhancing autophagy and regulating apoptosis pathways through Western blotting. Collectively, these data suggest that the potential mechanism of BA is a multi-target and multi-channel treatment for chlorpyrifos-induced liver injury.

## 1. Introduction

Chlorpyrifos (CPF) (Figure 1A) is an organophosphate insecticide and acaricide that has been extensively applied in agriculture globally; however, negative effects have also occurred with the increase in chlorpyrifos use [1]. Studies have suggested that only approximately 10% of chlorpyrifos is capable of killing insects and mites, while the rest basically remains in soil, lakes, and various ecological environments, bringing great damage to ecosystems [2,3]. Especially when chlorpyrifos is ingested by various means, it can cause great harm to human beings due to its heterophilic nature, mainly including neurotoxicity, hepatotoxicity, reproductive toxicity, cardiotoxicity, and liver dysfunction [4]. CPF is mainly metabolized by the liver in the body. CPF is converted by cytochrome P450-mediated oxidative desulfurization in liver cells to CPF-Oxon, a potent acetylcholinesterase (ACHE) inhibitor, which is the main metabolite of chlorpyrifos exerting toxic effects in vivo [5]. The toxic effects of CPF-Oxon on the liver are mainly reflected in liver inflammation, oxidative stress, and apoptosis, leading to liver damage, but the mechanism is not clear.

Autophagy is a vital process of translocation of intracellular substances in normal and diseased cells in eukaryotes. The process of autophagy is usually considered as the detachment of a bilayer membrane from the rough endoplasmic reticulum into the cytoplasm, which then envelops the proteins and organelles that need to be decomposed to form autophagosomes, followed by fusion with lysosomes to form autophagolysosomes, which are degraded by various enzymes in lysosomes to maintain intracellular homeostasis and the renewal of certain organelles [6]. Autophagy is usually categorized into non-selective and selective. Non-selective autophagy usually occurs when cells are in a starvation state, and the cell initiates the autophagy program to devour and degrade intracellular components in order to provide various nutrients needed for the cell to survive [7]. Selective autophagy is a cellular response to a variety of adverse stimuli from the outside world, and selective autophagy can mitigate the damage to the cell by removing the misfolded proteins, piled-up lipids, and damaged organelles inside the cell [8]. The most unique feature of autophagy is that the positive or negative effects of autophagy in different diseases are different. Currently, the study of autophagy in CPF-induced liver injury has not been reported.

Baicalin (BA) (Figure 1B) is a flavonoid with significant biological activities, which is the main component of the dried root of the traditional Chinese medicine *Scutellaria baicalensis* Georgi [9]. Studies have found that baicalin is a multi-target, multi-channel drug with antibacterial, antiviral, antitumor, hepatoprotective, and other pharmacological effects, among which baicalin’s hepatoprotective effects are the most significant [10,11,12]. Although the liver is the largest digestive and metabolic organ of the body, it is vulnerable to the destruction of various adverse factors including viruses and drugs which cause damage to its function and structure. Hepatitis B virus (HBV) is the most harmful of all hepatitis viruses to the liver, while BA exerts a unique killing effect. HNF 1 α and HNF 4 α are key factors in replication, RNA synthesis, and HBV protein production after HBV infects liver cells. BA can significantly downregulate the expression of HNF 1 α and HNF 4 α in time and dose, thereby exerting anti-HBV effects [13].For duck hepatitis A virus type 1 (DHAV-1)-infected duck liver cells, BA mainly restores the mitochondrial structural integrity of duck liver cells to maintain their function, while lowering the production of ROS, thereby effectively improving the harm caused by oxidative stress to liver cells [14]. SREBP 1c is an important regulator of liver fat formation induced by alcohol and a high-fat diet, and the vital way BA improves fatty liver is by reducing the formation of liver fat by inhibiting the activity of SREBP 1c and enhancing the liver’s ability to decompose fat by inhibiting the binding of SREBP1c-mediated PNPLA3 to ATGL [15,16]. The protective effect of BA on cholestatic liver injury and drug-induced liver injury is mainly through the regulation of oxidative stress, inflammatory response, and apoptosis of liver cells [17]. At present, liver cancer has become a major killer of human health. Studies have indicated that BA can specifically inhibit the expression of proto oncogenes, reduce the carcinogenesis rate of normal cells, block the cell cycle of cancer cells, and induce apoptosis of cancer cells. In addition, BA still plays a critical role in inhibiting cancer cell proliferation and metastasis and tumor angiogenesis and invasion [18]. Based on the significant hepatoprotective effect of baicalin, this study investigated the protective effects and mechanisms of CPF-induced AML12 cells injury and CPF-induced acute liver injury in mice.

## 2. Results

### 2.1. LD50 of CPF

The LD50 of CPF in AML12 cells was determined as the modeling concentration of the cell experiment. Due to the toxic effect of CPF, the survival of AML12 cells was in inverse proportion to the CPF content, and the cell survival was experimentally determined to be 50% at 50 μM CPF, and thus the LD50 of CPF = 50 μM (Figure 2A).

### 2.2. The Relationship between BA and CPF on the Survival Rate of AML12 Cells

In this experiment, AML12 cells were pre-protected for 24 h with BA solution at a concentration gradient and then injured with 50 μM CPF for 24 h. When the densities of BA were 600, 800, and 1000 μg/mL, the cell survival rates were 66.58%, 79.67%, and 91.95%, respectively. Therefore, the concentrations of BA in this study were 600, 800, and 1000 μg/mL for the experimental concentration in the subsequent experiments (Figure 2B).

### 2.3. Determination of Cellular Indicators in Cell Supernatants

When AML12 cells are injured by CPF, intracellular ALT, AST, LDH, and γ-GT components are released into the supernatant. After pretreatment with BA, the secretion of the above indicators in the supernatant was significantly reduced, implying that BA pretreatment alleviated CPF-induced AML12 cells injury (Figure 2C).

### 2.4. The Contents of Inflammatory Factors in the Supernatant of Each Experimental Group

ELISA results demonstrated that CPF caused an inflammatory response in AML12 cells; thus, IL-6, IL-1β, and TNF-α were released in large amounts from AML12 cells. After pretreatment with BA, the content of the above three inflammatory factors in the cell supernatant gradually returned to normal levels. IL-10 is a multifunctional regulator playing a vital role in the body’s immune response, tumor growth or control, and inflammatory response, especially playing a central role in the anti-inflammatory response [19]. Its content change is opposite to that of pro-inflammatory factors, which indicates that BA exerts a significant inhibitory effect on CPF-induced AML12 cell inflammatory response (Figure 2D).

### 2.5. Protective Effect of BA against Oxidative Stress in CPF-Induced AML12 Cells

Oxidative stress was stimulated when all kinds of internal and external factors cause an imbalance in the oxidative system in the liver. At this time, the chromosomes, organelles, and cell membranes in liver cells will be destroyed, resulting in abnormal liver structure and function [20]. As a result, the detection of liver oxidative stress markers can diagnose liver injury and determine the potential of drug therapy. The significant increase in MDA content in the model group showed that CPF-induced AML12 cells produced a large number of superoxide free radicals, promoted the superoxide oxidation of cell membrane lipids, and destroyed the structure and function of AML12 cells. At the same time, the contents of related enzymes including SOD, CAT, and GSH-PX, which were antagonistic to oxidative stress, were much lower than normal levels, indicating that CPF induced oxidative stress in AML12 cells. After pretreatment with BA, the above indicators gradually returned to normal levels, suggesting that BA can significantly improve CPF-induced oxidative stress in AML12 cells (Figure 2E).

### 2.6. Annexin V-FITC/PI Staining

The above experiments have demonstrated that CPF makes a pro-apoptotic effect on AML12 cells while BA has a protective effect on this apoptosis. We examined the differences in apoptosis in each experimental group using flow cytometry. The apoptosis rate of the model group was 52.36%, and the apoptosis rates of the 600, 800, 1000 μg/mL experimental groups were 39.73%, 28.60%, and 14.20%, respectively. The differences in apoptotic cells in the above experimental groups demonstrated that BA has a protective effect on the necroptosis of CPF-induced AML12 cells (Figure 3A).

### 2.7. BA Improves CPF-Induced AML12 Cells Injury by Enhancing Autophagy

The results of transmission electron microscopy showed the difference of autophagosome expression in the experimental groups, suggesting that BA could improve the inhibition of autophagy expression in AML12 cells by CPF (Figure 3B). The expression of LC3II protein, which is necessary for autophagosome formation, and Atg7 protein promotes autophagosome growth, better explaining the role of autophagy in the treatment of CPF-induced AML12 cell damage by BA. The sharp decrease in LC3II and Atg7 proteins after CPF induction and the sharp increase after BA pre-protection are consistent with the above experimental results. In addition, it can also be demonstrated that the expression of P62, which is inversely proportional to the activity of autophagy, in the model group and the administration group is just in the opposite to LC3 II and Atg7, proving that autophagy plays a positive role in BA improving CPF-induced AML12 cells injury (Figure 4A).

### 2.8. Study on Mechanism of Apoptosis of CPF-Induced AML12 Cells Induced by BA

The Bcl-2/Bax/Caspase-3 signaling pathway is an important pathway to regulate apoptosis. When the cells are under different conditions, the expression of Bax and Bcl-2 is out of balance, with apoptosis increasing or decreasing to maintain the normal operation of cells [21]. CPF adjusted the expression of Bax and Bcl-2 in AML12 cells, and increased Bax/Bax homologous dimers. Meanwhile, large amounts of Cyt-c activated the downstream apoptotic signaling molecule cleaved caspase-3, which enabled AML12 cells to initiate pro-apoptotic procedures. In the case of BA pre-protection, Bax/ Bcl-2 allodimer increased, blocking Cyt-c release, reducing cleaved caspase-3 activity, and enabling AML12 cells to initiate anti-apoptotic procedures (Figure 4B).

### 2.9. BA Improved CPF-Induced Kunming Female Mouse Liver Injury

In this experiment, we usually evaluate the protective effect of BA against in vivo liver injury by measuring various liver function indicators in the serum of a Kunming female mouse, from which substances including ALT, AST, LDH, and γ-GT are ruptured. When BA is pre-protected, the amount of disruption of liver cells is reduced, and the abnormal release of the above four indicators is improved (Figure 5A).

### 2.10. BA Improved the Inflammatory Response of CPF-Induced Kunming Female Mouse Liver Injury

Liver injury is usually accompanied by liver inflammation, and thus we can evaluate the grade of liver injury by measuring the severity of liver inflammation [21]. The toxic effect of CPF causes an inflammatory response in the liver, and three pro-inflammatory factors such as IL-6 are released in large quantities. After pretreatment with BA, the above three indexes were effectively decreased, while the change trend of anti-inflammatory factors was just in the opposite, which indicated that BA could effectively protect CPF-induced Kunming female mouse liver injury (Figure 5B).

### 2.11. BA Improved Oxidative Stress of CPF-Induced Kunming Female Mouse Liver Injury

Liver injury is the foundation for the occurrence and development of all liver diseases, and oxidative stress can promote this process [22]. Compared with the normal content, CPF stimulation significantly increased the MDA content in mouse liver cells. At the same time, the decrease in the content of antioxidant enzymes such as SOD reveals a decrease in the antioxidant capacity of the liver. When BA was pre-protected, the MDA content in the liver of mice was effectively inhibited, and the contents of SOD, CAT, and GSH-PX gradually recovered (Figure 5C).

### 2.12. Liver Histopathology Testing

We detected the histopathological status of mouse livers by HE and Masson staining. Normal liver cells were neatly arranged, cytoplasm was rich, and no obvious histopathological changes were observed. CPF caused liver cell disorders and hepatocellular edema in mice, which was manifested as balloon-like degeneration. Hepatocyte necrosis was accompanied by inflammatory cell infiltration. In the case of BA pre-protection, the pathological changes of edema, necrosis, and inflammatory cell infiltration of liver cells gradually improved (black short-tail arrow hepatocyte edema; black long arrow-inflammatory cell infiltration; black liver cell necrosis) (Figure 6A). Masson results indicated that there was no obvious fibrous tissue hyperplasia in the control group. However, the fibrous portal area was widened. The sinus fibrosis was degenerated and the formation of bridging fibrosis was observed when CPF was given. When given BA pre-protection, the degree of liver fibrosis in each experimental group gradually improved (black short-tailed arrow collagen fibers) (Figure 6B).

### 2.13. TUNEL Staining Analysis

In this study, the increasing number of green fluorescence cells showed that CPF promoted increased apoptosis in living liver cells in mice. When BA is pre-protected, the number of green fluorescence cells gradually decreases, indicating that the pre-protective effect of BA plays a role in inhibiting abnormal apoptosis of mouse liver cells (Figure 6C).

### 2.14. BA Exerts Anti-CPF-Induced Kunming Female Mouse Liver Injury by Enhancing Autophagy

To investigate the protective mechanism of BA on CPF-induced Kunming female mouse liver injury, we measured LC3II/I, Atg7, and P62 in the liver. After CPF gavage, the conversion rate of LC3II/I and the production of Atg7 were decreased, and the production of P62 was increased. After pretreatment with BA, the conversion rate of LC3II/I and the production of Atg7 were significantly increased, and the production of P62 was significantly reduced, indicating that more autophagosomes were synthesized in the administration group (Figure 7A).

### 2.15. BA Improved CPF-Induced Kunming Female Mouse Liver Injury by Regulating Apoptosis Pathway

The accumulation of CPF in the liver of mice increased the expression of Bax in liver cells, formed more Bax/Bax homodimers, and enhanced the response of liver cells to apoptosis signal. In the meanwhile, the release of Cyt-c increased the expression of cleaved caspase-3 correspondingly, promoting apoptosis of liver cells. In the case of BA pre-protection, Bcl-2 expression increases, Bax/Bcl-2 heterodimer increases, the release of Cyt-c and the synthesis of cleaved caspase-3 decreases, and liver cells initiate anti-apoptotic procedures (Figure 7B).

## 3. Discussion

CPF is a new type of organophosphate pesticide; its emergence overcomes the shortcomings of octathion, dichlorvos, and other organophosphate pesticides to make it the world’s largest organophosphate pesticide. Moreover, it is also widely used in various agricultural production and pest control ventures that are harmful to animals and the human living environment [23]. However, with the increase in use and the extension of the use time, it has been found that some workers who are often exposed to CPF have a higher proportion of dizziness, nausea, chills, confusion, and other symptoms; coincidentally, in the United States, some families that often use CPF as a pesticide have a higher rate of infant malformations, and during the process of later growth, they show serious deficiencies in intelligence, immunity, and reproduction [24]. These phenomena have forced people to start to examine the safety of CPF, and after numerous examples and experiments, it is known that CPF can enter the human body through the gastrointestinal tract, respiratory system, and skin and cause serious harm to various organs and systems of the human body [25,26].

At present, it is known that the harm of CPF to the human body is mainly concentrated in the following aspects. Firstly, CPF, as a cholinesterase inhibitor, exhibits significant neurotoxicity [27]. Secondly, CPF can cause serious damage to the human reproductive system. From the perspective of women, CPF can cause apoptosis of stromal cells through the placenta during pregnancy, change the composition of the villi matrix, destroy the thickness of the basement membrane and the integrity of the trophoblast, as well as cause serious consequences including fetal malformation and miscarriage [28,29]. From a male perspective, CPF can significantly reduce the weight of the testicles and epididymis, cause seminiferous tubular atrophy and degeneration, reduce the number of spermatozoa, the exercise capacity is reduced and the occurrence of malpractice, plasma testosterone, follicle-stimulating hormone, luteinizing hormone, and testosterone steroid content is significantly reduced. In Vietnam, it has been reported that the DNA of some people exposed to CPF for a long time has been damaged, causing genotoxicity [30]. However, the most serious thing is that the metabolism of CPF occurs in the liver and the blood flow in the liver is extremely rich, making CPF cause great damage to the liver, while the current research on the prevention and treatment of liver damage caused by CPF by traditional Chinese medicine compounds is seriously insufficient [29].

In this experiment, we investigated the protective effect and mechanism of BA on CPF-induced liver injury from both in vitro and in vivo perspectives using a CPF-induced mouse normal liver cell AML12 cells damage model and a CPF-induced liver injury model in Kunming female mice. Normally, substances such as ALT, AST, LDH, and γ-GT are present in liver cells. However, when liver cells are damaged, the permeability of cell membranes is enhanced, and these substances are released from liver cells into the cell supernatant of the latter blood, especially ALT and AST, which are two enzymes unique to catalyzing transamination reactions in liver cells and play a vital role in the process of liver physiology [30]. Therefore, measuring the amount of these substances in the cell supernatant or blood can tell if liver cells have been damaged. In this study, we investigated the protective effect of BA on CPF-induced liver injury at both in vitro and in vivo levels. When AML12 cells and liver were exposed to CPF, the contents of ALT, AST, LDH, and γ-GT in the cell supernatant and serum of mice were significantly increased, indicating that CPF caused significant damage to AML12 cells and the liver. When BA was given pre-protection, the content of the above substances in the cell supernatant and mouse serum was significantly decreased, which indicated that BA had a significant protective effect on the liver injury caused by CPF. When liver cells are destroyed, inflammatory response and oxidative stress occur, which are the pathological basis for the occurrence and development of all liver diseases [31].

In the inflammatory response, IL-6 is the primary pro-inflammatory factor in the inflammatory response of the liver, and the large production of IL-6 induces the production of subsequent inflammatory factors and a series of related physiological responses [32]. Fundamentally, the body’s inflammatory response is an autoimmune mechanism in the face of external stimuli, and IL-1β exerts a vital role in the immune process and naturally participates in the inflammatory response [33]. IL-1β plays a role in the whole inflammatory response, and when IL-1β binds to downstream signaling molecules, it can not only promote the release of cytokines such as TNF-α, but also accelerate apoptosis. TNF-α is a key factor in promoting apoptosis [34]. In the inflammatory response, the main role of the anti-inflammatory factor IL-10 is to lower the production of inflammatory factors through macrophages [35]. In this study, we explored the protective effect of BA on CPF-induced liver injury inflammation at both in vitro and in vivo levels. When AML12 cells and liver were exposed to CPF, the contents of IL-6, IL-1β, and TNF-α in the cell supernatant and serum of mice were significantly increased, and the content of anti-inflammatory factor IL-10 was significantly decreased, indicating that CPF caused severe inflammation in both AML12 cells and the liver. In the case of BA pre-protection, the content of the above proinflammatory factors was significantly reduced. On the contrary, the content of the anti-inflammatory factor IL-10 was significantly increased, which indicated that BA had a significant protective effect on the inflammatory response of CPF-induced liver injury.

MDA is the most representative biological indicator used to measure the oxidation of cells, and its content can reflect the degree of cell damage. SOD, CAT, and GSH-PX play a role in scavenging free radicals and antioxidants after oxidative stress in the body [36]. This study investigated the effect of BA on oxidative stress induced by CPF-induced liver injury at both in vitro and in vivo levels. When AML12 cells and liver were exposed to CPF, MDA content in AML12 cells and liver was significantly increased, while SOD, CAT and GSH-PX contents were significantly decreased, demonstrating that CPF stimulated oxidative stress in AML12 cells and the liver. Under the condition of BA pre-protection, MDA content in AML12 cells and liver was significantly reduced, while SOD, CAT, and GSH-PX contents were significantly increased. This indicated that BA had a significant improvement effect on CPF-induced oxidative stress in the liver.

Apoptosis is a complex process regulated by a variety of factors, among which the Bcl-2/Bax/Caspase-3 signaling pathway is a widely recognized classical apoptosis pathway. When the expression of Bax is activated by internal and external factors, cells enter the apoptotic process, and when the expression of Bcl-2 is activated, cells enter the anti-apoptotic process [37,38]. This study explored the protective effect and mechanism of BA on CPF-induced apoptosis in vivo and in vitro. Flow cytometry and TUNEL staining were employed to detect the apoptosis rate. When AML12 cells and liver were exposed to CPF, the apoptosis rate of AML12 cells and liver cells in vivo increased significantly. When BA was given pre-protection, the apoptosis rate was significantly reduced in vitro and in vivo. Regarding mechanism, when AML12 cells and liver were exposed to CPF, the expressions of Bax and cleaved caspase-3 were significantly increased, and the expression of Bcl-2 was significantly decreased, initiating the apoptosis process. In the case of BA pre-protection, the expression levels of Bax and cleaved caspase-3 were significantly decreased, and the expression level of Bcl-2 was significantly increased, implying that BA had a protective effect on CPF-induced apoptosis through regulation of the Bcl-2/Bax/Caspase-3 pathway. Finally, this study also found that BA enhanced the protective effect of autophagy on CPF-induced liver injury.

## 4. Materials and Methods

### 4.1. Experimental Articles and Detection Reagents

CPF was provided by Kaifeng Wave Chemical Co., Ltd., Kaifeng, Henan, China. CPF was diluted with DMEM/F12 into various concentration gradients for use. Saline was purchased from Tonghua Dongbao Pharmaceutical Co., Ltd. (Tonghua, China). Baicalin (purity ≥ 95%) was purchased from Chengdu Pfizer Biotechnology Co., Ltd. (Chengdu, China). Baicalin was dissolved to a specific final concentration prior to use.

### 4.2. Cell Culture

Mouse normal hepatocytes AML12 cells used in in vitro experiments were provided by the Chinese Academy of Sciences Cell Bank. AML12 cells were cultured with DMEM/F12 medium containing 10% FBS and 1% ITS in a T75 cell culture flask. The cells were digested with 1× trypsin when they reached 90% of their length and then digested at 1 × 10^6^ cells/mL (100 μL per well) within the 96-well plates and cultured for 10 h to reach a stable state for the subsequent experiments.

### 4.3. CCK-8 Measures the LD50 of CPF on AML12

The medium was aspirated from the 96-well plate and washed with PBS. CPF 0, 10, 20, 30, 40, 50, 60, 70, 80, 90, and 100 μM were added per row for 24 h and incubated in each well for 30 min. Finally, the OD value was determined using a microplate reader (Varioskan: Thermo Scientific, Wuhan, China) in the absence of light to calculate the toxic effect of CPF.

### 4.4. Screening of Baicalin Therapeutic Concentrations and AML12 Cells Treatment

The well-grown cells were washed three times with PBS according to 1 × 10^6^ cells/mL (100 μL per well) within the 96-well plates and cultured for 10 h to remove the medium. The cells were treated with baicalin at a concentration of 200, 400, 600, 800, 1000, 1200, 1400, 1600, 1800, and 2000 μg/ mL for 24 h. Then, the baicalin solution was removed and the LD50 of CPF was added. After 24 h, 10 μL CCK-8 was placed in each well and incubated at 37 °C for 30 min. Cell viability was measured using microplate reader (Varioskan: Thermo Scientific). The cell experiment was divided into control group, model group, and treatment groups. The incubated AML12 cells treated with 0 μM CPF and 0 μg BA were named the control group. In the treatment groups, three concentrations were selected from the above treatment concentrations and considered as low, medium, and high doses, respectively (Figure 3A).

### 4.5. ELISA

According to 1 × 10^6^ cells/mL (2 mL per well) within the 6-well plates, the CPF-induced AML12 injury model was established in line with the LD50 of CPF and the effective concentration of baicalin selected in the above experiments. The cell supernatant of each experimental group was extracted based on the concentration at 1000× *g* for 5 min. The cell supernatant was extracted according to ELISA kit instructions.

### 4.6. Measurement of Cellular Oxidative Stress Indicators

After cell culture and experiment, the medium in 6-well plates was aspirated off and washed with PBS three times, followed by digestion with 1× trypsin. The cells were resuspended using PBS, and centrifuged in EP tubes. The supernatant was aspirated and the indicators were measured according to the instructions.

### 4.7. Annexin V-FITC/PI Staining

The medium in the 6-well plates was sucked into the centrifuge tube for use. After the digestion of the cells, the cells are beaten down using the previously collected medium, collected in centrifuge tubes, and then centrifuged to discard the supernatant. Cells are washed using PBS and 1× binding buffer. Then, 5 μL LAnnexin V and 5 μLPI were added to 100 uL of cell suspension (1 × 10^5^ cells/mL), incubated at room temperature protected from light, and subsequently detected after 1:4 addition of 1× binding buffer.

### 4.8. TEM

After the end of the experiment, the cells were washed three times with PBS. After trypsin digestion, the cells were transferred to the EP tube with 500 μL of medium and centrifuged. After discarding the supernatant, 200 μL of 2.5% glutaraldehyde was added and the cells were fixed overnight at 4 °C. PBS was washed three times, with 4 °C precooling 1% osmic acid fixed for 2 h. PBS was washed three times, using the volume fraction of 0.5, 0.7, 0.8, 0.9, and 1 of ethanol after using the volume fraction of 1 acetone gradient dehydration, epoxy resin infiltration, and embedding after using an ultrathin slicer cut into 50~70 nm thin slices. Double staining of uranium acetate-lead citrate was observed and photographed by transmission electron microscopy.

### 4.9. Animal Experimental Program

In this experiment, 40 SPF female Kunming mice were obtained from Beijing Huafukang Biotechnology Co., Ltd. (Beijing, China), which were aged 10 weeks, weighing (35 ± 2) g. Experimental animals were kept in the animal experiment center on the fourth floor of the Science and Technology Innovation Center of Jilin Agricultural University. During the experiment, the mice were free to eat and drink water, which were maintained at room temperature (25.0 ± 2.0) °C and humidity (60 ± 5)%. The light/dark cycle was carried out for 12 h. The formal test began after one week of adaptive feeding. Forty mice were randomly divided into five groups. One group received a conventional oral gavage with normal saline as the control group (marked as control). Based on the previous studies of our research group, 10 mg/kg/d CPF was selected as the modeling concentration (marked as model), with 50, 100, and 200 mg/kg/d being selected as the experimental doses (marked as L-baicalin, M-baicalin, H-baicalin). Pre-protective administration was used in the experiment, and CPF was administered 1 h after intragastric administration of baicalin (Figure 8A). During the experiment, the growth status of mice was observed every day, and the weight changes of mice were recorded (Figure 8B).

### 4.10. Measurement of Liver Function and Inflammatory Factors

At the end of the experiment, blood was collected from the retro-ocular venous plexus and centrifuged at 1000× *g* for 5 min to absorb the serum. In addition, the liver function indexes and inflammatory factors in serum were detected by ELISA kit (Nanjing Jiancheng Bioengineering Institute, Nanjing, China).

### 4.11. Measurement of Oxidative Stress Indicators

Mice are dissected after sacrifice and liver tissue is cut and weighed for the subsequent experiments. The normal saline and liver tissue were cut into pieces at a ratio of 1:9 and broken by ultrasonic disruptor until the liver tissue was homogenized. The supernatant was centrifuged at 1000× *g* for 5 min to determine the contents of oxidative stress indexes in line with the manufacturer’s instructions.

### 4.12. Liver Histopathology Testing

Fresh liver tissue was fixed in 4% paraformaldehyde for more than 24 h. After dehydration with anhydrous ethanol and paraffin embedding, the wax blocks were cut into 4μm sections. After dehydration, the sections were stained with HE and Masson stain and subsequently analyzed by microscopic examination.

### 4.13. TUNEL Staining

The paraffin sections of the liver were dewaxed according to the conventional method, followed by repair, membrane rupture, TUNEL staining, DAPI staining, sealing, and other operations. Finally, the sections were placed under a microscope in order to observe apoptosis and the nucleus.

### 4.14. Western Blot

AML12 cells and mouse liver tissues were grinded separately to extract proteins. After protein concentration determination, primary and secondary antibodies were incubated by SDS-PAGE electrophoresis, membrane transfer, blocking, and successive incubation. Membrane washing, development, and other operations were performed to determine the expression of LC3, Atg7, P62, Bax, Bcl-2 and cleaved caspase-3.

### 4.15. Statistical Analysis

In this study, the Image J-win64 software processing system was employed to process the grayscale value of Western blotting. GraphPad Prism 8.0 statistical software and the analysis of variance (ANOVA) method were used to analyze and process all data. The data were represented as “average ± standard deviation”.

## 5. Conclusions

The present study demonstrates that baicalin can significantly attenuate CPF-induced liver injury by inhibiting inflammatory response, oxidative stress, and cell apoptosis in vivo and in vitro. Furthermore, we provide an insight into the mechanisms underlying the CPF-induced liver injury protective role of baicalin by regulating autophagy and apoptosis pathways. Moreover, our study is of great significance for liver toxicity studies of CPF and provides scientific basis for the use of baicalin in the treatment of liver injury induced by CPF in the future.

## Figures and Tables

**Figure 1 molecules-28-07771-f001:**
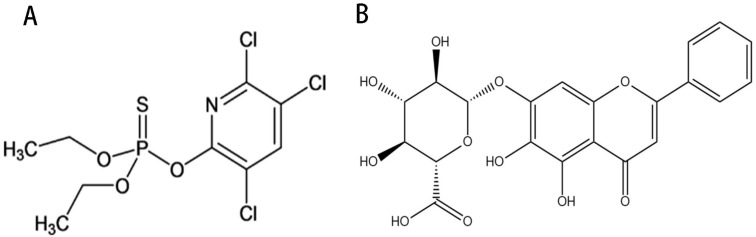
Compound structures of (**A**) chlorpyrifos, (**B**) baicalin.

**Figure 2 molecules-28-07771-f002:**
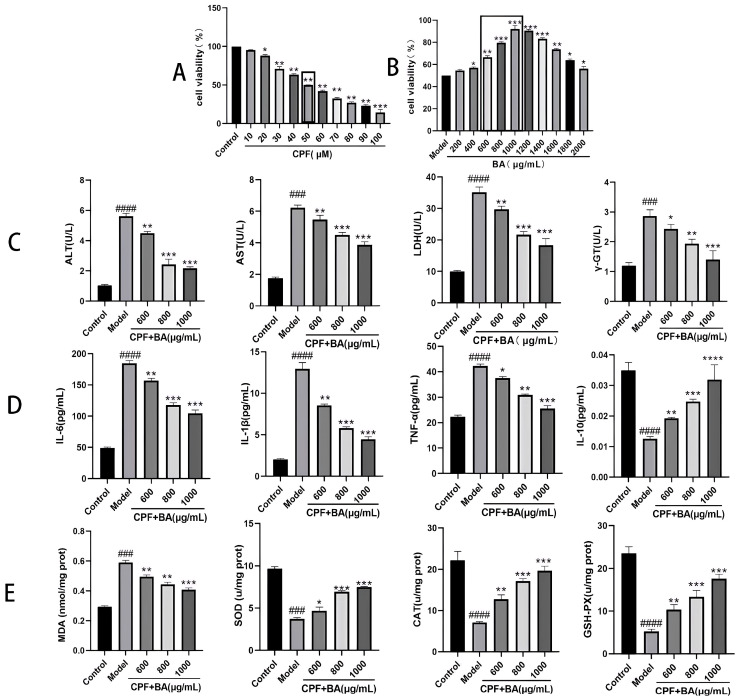
Establishment of cell model and determination of various indexes. (**A**) The relationship between CPF content and survival rate of AML12 cells. (**B**) The relationship between BA and the survival rate of CPF-induced in AML12 cells. (**C**) The contents of ALT, AST, LDH, and γ-GT in the cell supernatant of each experimental group. (**D**) The contents of inflammatory factors in the supernatant of each experimental group. (**E**) The content of oxidative stress indexes in the cells of each experimental group; ### *p* < 0.001, #### *p* < 0.0001 vs. Control group; * *p* < 0.05, ** *p* < 0.01, *** *p* < 0.001, **** *p* < 0.0001 vs. Model group (*n* = 3).

**Figure 3 molecules-28-07771-f003:**
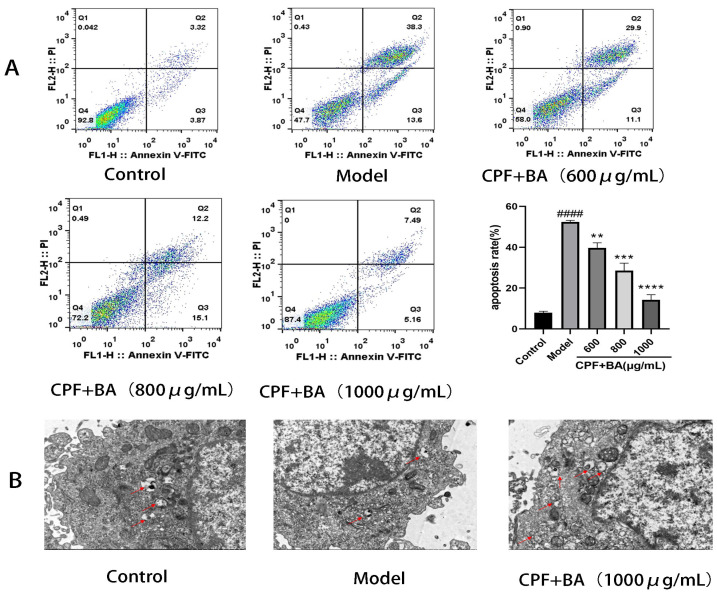
Flow cytometry and TEM. (**A**) Flow cytometry. (**B**) TEM. #### *p* < 0.0001 vs. Control group; ** *p* < 0.01, *** *p* < 0.001, **** *p* < 0.0001 vs. Model group.

**Figure 4 molecules-28-07771-f004:**
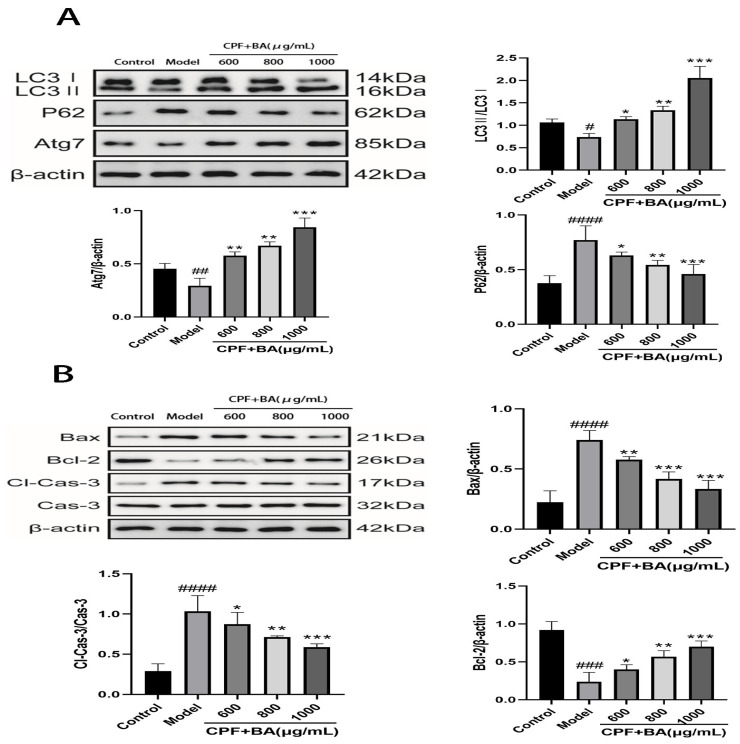
The mechanisms of BA on CPF-induced AML12 cells injury. (**A**) The effect of BA on autophagy pathway. (**B**) The effect of BA on Bcl-2 /Bax/Caspase-3 signaling pathway. # *p* < 0.001, ## *p* < 0.001, ### *p* < 0.001, #### *p* < 0.0001 vs. Control group; * *p* < 0.05, ** *p* < 0.01, *** *p* < 0.001,vs. Model group.

**Figure 5 molecules-28-07771-f005:**
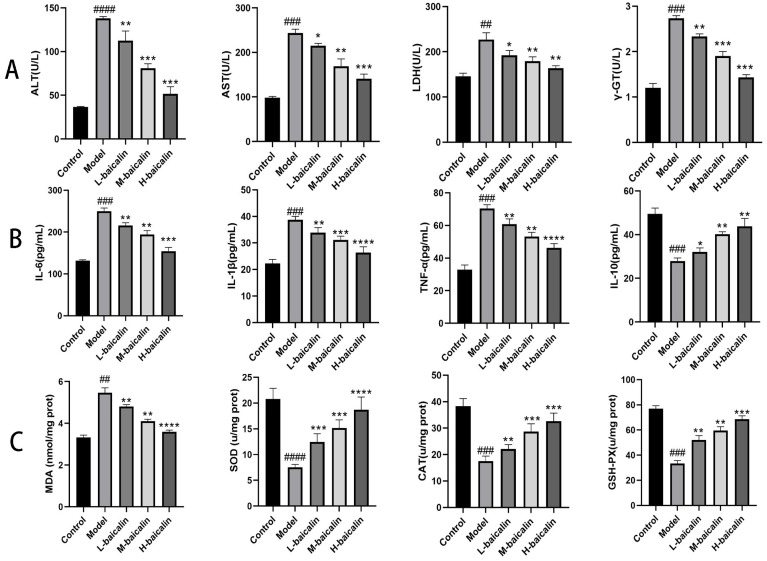
Determination of BA on liver function, inflammatory factors, and oxidative stress in CPF-induced Kunming female mouse. (**A**) Effect of BA on liver function in CPF-induced Kunming female mouse. (**B**) Effect of BA on liver function in CPF-induced inflammatory factors in Kunming female mouse. (**C**) Effect of BA on oxidative stress indicators of BA on CPF-induced inflammatory factors in Kunming female mouse. ## *p* < 0.001, ### *p* < 0.001, #### *p* < 0.0001 vs. Control group; * *p* < 0.05, ** *p* < 0.01, *** *p* < 0.001, **** *p* < 0.0001 vs. Model group (*n* = 3).

**Figure 6 molecules-28-07771-f006:**
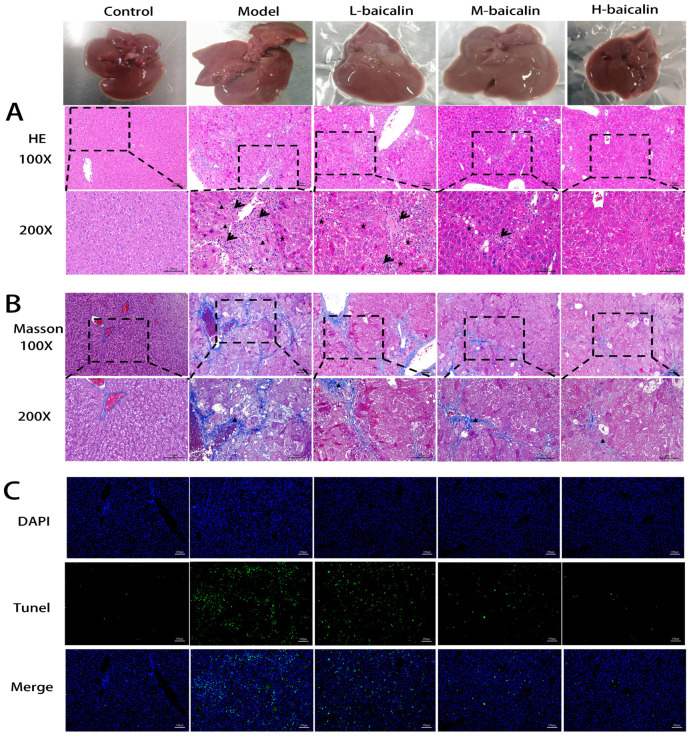
Effect of BA on liver fibrosis testing and TUNEL stain in CPF-induced Kunming female mouse liver injury. (**A**) HE stain (Triangle, arrow and asterisk represent hepatocyte edema, inflammatory cell infiltration and hepatocyte necrosis, respectively) (Top: scale bar 100 μm, bottom: scale bar 200 μm). (**B**) Masson stain (Top: scale bar 100 μm, bottom: scale bar 200 μm). (**C**) TUNEL stain (scale bar 100 μm).

**Figure 7 molecules-28-07771-f007:**
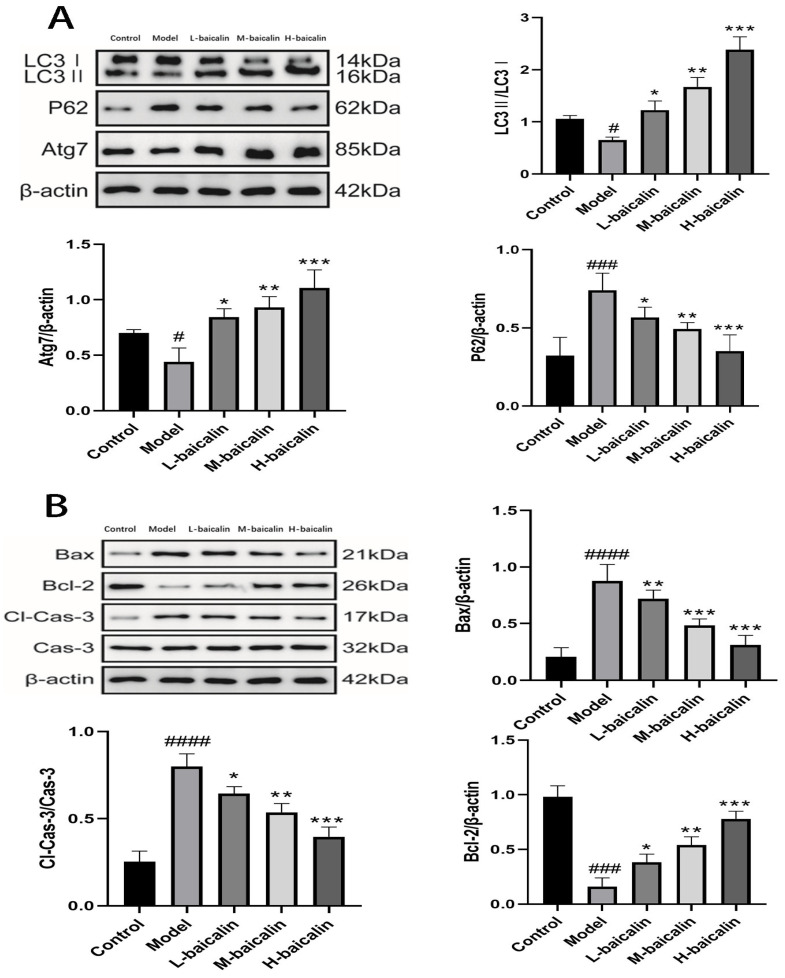
The mechanism of BA on CPF-induced Kunming female mouse liver injury. (**A**) The effect of BA on autophagy pathway. (**B**) The effect of BA on apoptosis pathway. # *p* < 0.001, ### *p* < 0.001, #### *p* < 0.0001 vs. Control group; * *p* < 0.05, ** *p* < 0.01, *** *p* < 0.001 vs. Model group.

**Figure 8 molecules-28-07771-f008:**
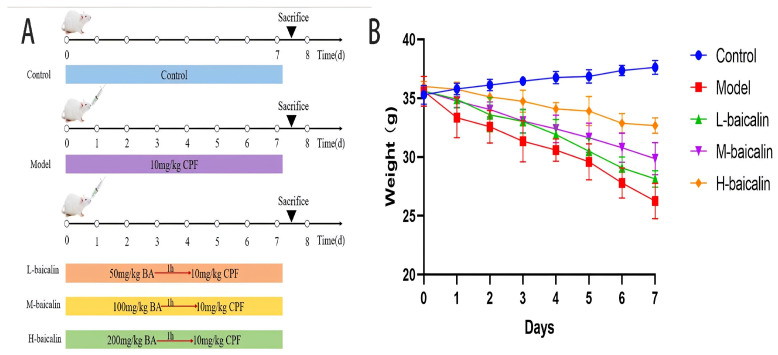
(**A**) Animal experimental program (*n* = 8). (**B**) Weight change curve of mice in each experimental group (*n* = 8).

## Data Availability

The original data of this paper have been shared to the authors within a reasonable demand.

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
