# Peer review of "Protective Effect of Baicalin on Chlorpyrifos-Induced Liver Injury and Its Mechanism"

_molecules, 2023, doi:10.3390/molecules28237771_

Round 1

Reviewer 1 Report

Comments and Suggestions for Authors

molecules-2659979

Protective effect of baicalin on chlorpyrifos-induced liver injury and its mechanism

Some suggestions are given below.

1-      In the abstract the concentrations at which BA could significantly reduce the inflammatory factors, various oxidative stress indicators and apoptosis in the supernatant of CPF-induced AML12 cells injury should be mentioned.

Be more specific regarding previous reports of hepatoprotective flavonoids. Incorporate a paragraph commenting on them.

2-      In the methodology of the different tests, the controls used should be mentioned more specifically.

3-      Discussion:

The work discussion should be reviewed, 90% is referring to liver injury and background (lines 239-232). It is not understood why all that information is included. .

The discussion should be rewritten and based on the results of the manuscript.

The discussion should be reviewed in depth, as it is presented it is not suitable for the Journal.

Comments on the Quality of English Language

Moderate editing of English language required

Author Response

Response to reviewer 1

Thank you for your positive and helpful comments. We address the comments as follows.

Point 1: In the abstract the concentrations at which BA could significantly reduce the inflammatory factors, various oxidative stress indicators and apoptosis in the supernatant of CPF-induced AML12 cells injury should be mentioned. Be more specific regarding previous reports of hepatoprotective flavonoids. Incorporate a paragraph commenting on them.

Response 1:

Thank you for your comments. We have added a description of “BA could significantly reduce the inflammatory factors, various oxidative stress indicators and apoptosis in the supernatant of CPF-induced AML12 cells” in our abstract. Moreover, we added the previous reports of hepatoprotective mechanism of baicalin in the introduction part(line57-line81).

Point 2:  In the methodology of the different tests, the controls used should be mentioned more specifically.

Response 2:

Thank you for your comments. The controls have been mentioned specifically

both in cell and animal tests(lines 373-377,413-418).

Point3:  The work discussion should be reviewed, 90% is referring to liver injury and background (lines 239-232). It is not understood why all that information is included. The discussion should be rewritten and based on the results of the manuscript.

The discussion should be reviewed in depth, as it is presented it is not suitable for the Journal.

Response 3:

Thanks for this comment. We have rewritten the discussion section based on our own research findings.

Reviewer 2 Report

Comments and Suggestions for Authors

Thank you for the article. I have these comments;

- Please improve the resolution of figures.

- Did the authors check the toxicity of this flavonoid compound on normal cell lines?

- I am worrying about the stability of this flavonoid compound in the different media used in in vitro and in vivo assays. The flavonoid contains a sugar and a carboxylic acid which may be changed somehow in the assay. Could you please check by HPLC the stability of the compound when you first add in the media and after finishing the experiment? 

- In general, the pharmacokinetic profile of flavonoids, is not so good to promote their use/application as true drugs, based on that, please provide a clear aim and application of your finding?

Comments on the Quality of English Language

Moderate editing of English language required

Author Response

Thank you for your positive and helpful comments. We address the comments as follows.

Point 1: Please improve the resolution of figures.

Response 1:Thanks for this comment. We have already improved the resolution of all figures in our article.

Point 2:Did the authors check the toxicity of this flavonoid compound on normal cell lines?

Thank you for your comments. We used CCK-8 to determine the toxic effect of baicalin on AML12 normal cell lines, and the experimental results showed that baicalin was present at 0 μg/mL-1000 μg/mL has no toxic effect on AML12 normal cell lines. The experimental results are shown in Excel (Table 1)and the picture(Figure 1).

Point 3: I am worrying about the stability of this flavonoid compound in the different media used in in vitro and in vivo assays. The flavonoid contains a sugar and a carboxylic acid which may be changed somehow in the assay. Could you please check by HPLC the stability of the compound when you first add in the media and after finishing the experiment? 

Response3:

    The stability of the baicalin has been checked by HPLC. Our results indicated that baicalin is stable in DMEM medium and the content remains basically unchanged(Figure 2). Other research results showed that a mutual transformation of baicalin and its aglycone, baicalein occurred in the body once baicalin was absorbed. Indeed, baicalin was hydrolyzed to baicalein by β-glucuronidase in the intestine soon after administration, and then, baicalein was transformed back to baicalin by UDP-glucuronosyltransferase once it entered the systemic circulation [1]. Thus, baicalin is the major component in the systemic circulation rather than baicalein [2].

[1] Noh K, Kang Y, Nepal M, Jeong K, Oh D, Kang M, et al. Role of intestinal microbiota in baicalin-induced drug interaction and its pharmacokinetics. Molecules. 2016;21(3):337. https ://doi.org/10.3390/molec ules2 10303 37.

[2] Li M, Shi A, Pang HX, Xue W, Li Y, Cao GY, et al. Safety, tolerability, and pharmacokinetics of a single ascending dose of baicalein chewable tablets in healthy subjects. J Ethnopharmacol. 2014;156: 210–5.

https ://doi.org/10.1016/j.jep.2014.08.031.

Point 4: In general, the pharmacokinetic profile of flavonoids, is not so good to promote their use/application as true drugs, based on that, please provide a clear aim and application of your finding?

Response 4:

Thank you for the comments. Baicalin, as an oral preparation which named Baicalin Capsules, has got the code number H20158009 approved by China Food and Drug Administration. This drug is used in clinical practice to treat acute and chronic hepatitis. The relevant information of this drug could be found at https://www.baiji.com.cn/goods-15494.html. At present, there is no clear medication in clinical practice that can treat chlorpyrifos poisoning. Our research has demonstrated that baicalin could treat liver injury induced by chlorpyrifos, which could provide important reference for the clinical use of Baicalin Capsules to treat chlorpyrifos poisoning. Therefore, our study is very meaningful.

Round 2

Reviewer 1 Report

Comments and Suggestions for Authors

The authors have responded to the suggestions made. The paper should be accepted in its current state

Comments on the Quality of English Language

Minor editing of English language required